# Structure of cyanobacterial photosystem I complexed with ferredoxin at 1.97 Å resolution

Jiannan Li[1,2], Noriyuki Hamaoka[1,3], Fumiaki Makino [4,5], Akihiro Kawamoto [1], Yuxi Lin[6], Matthias Rögner [7], Marc M. Nowaczyk [7], Young-Ho Lee [6,8,9], Keiichi Namba [4,10,11], Christoph Gerle [1,12✉] & Genji Kurisu [1,2,3,13✉]

Photosystem I (PSI) is a light driven electron pump transferring electrons from Cytochrome $c_6$ (Cyt $c_6$) to Ferredoxin (Fd). An understanding of this electron transfer process is hampered by a paucity of structural detail concerning PSI:Fd interface and the possible binding sites of Cyt $c_6$. Here we describe the high resolution cryo-EM structure of *Thermosynechococcus elongatus* BP-1 PSI in complex with Fd and a loosely bound Cyt $c_6$. Side chain interactions at the PSI:Fd interface including bridging water molecules are visualized in detail. The structure explains the properties of mutants of PsaE and PsaC that affect kinetics of Fd binding and suggests a molecular switch for the dissociation of Fd upon reduction. Calorimetry-based thermodynamic analyses confirms a single binding site for Fd and demonstrates that PSI:Fd complexation is purely driven by entropy. A possible reaction cycle for the efficient transfer of electrons from Cyt $c_6$ to Fd via PSI is proposed.

[1] Laboratory for Protein Crystallography, Institute for Protein Research, Osaka University, Suita, Osaka 565-0871, Japan. [2] Department of Biological Sciences, Graduate School of Science, Osaka University, Toyonaka, Osaka 560-0043, Japan. [3] Department of Macromolecular Science, Graduate School of Science, Osaka University, Toyonaka, Osaka 560-0043, Japan. [4] Graduate School of Frontier Biosciences, Osaka University, Suita, Osaka 565-0871, Japan. [5] JEOL Ltd., Akishima, Tokyo, Japan. [6] Research Center for Bioconvergence Analysis, Korea Basic Science Institute, Ochang, Chungbuk 28119, South Korea. [7] Plant Biochemistry, Faculty of Biology and Biotechnology, Ruhr University Bochum, 44780 Bochum, Germany. [8] Bio-Analytical Science, University of Science and Technology, Daejeon 34113, South Korea. [9] Graduate School of Analytical Science and Technology, Chungnam National University, Daejeon 34134, South Korea. [10] RIKEN Center for Biosystems Dynamics Research and SPring-8 Center, Suita, Osaka, Japan. [11] JEOL YOKOGUCHI Research Alliance Laboratories, Osaka University, Suita, Osaka, Japan. [12] RIKEN SPring-8 Center, Kouto, Sayo-gun, Hyogo 679-5198, Japan. [13] Institute for Open and Transdisciplinary Research Initiatives (OTRI), Osaka University, Suita, Osaka 565-0871, Japan. ✉email: christoph.gerle@riken.jp; gkurisu@protein.osaka-u.ac.jp

Oxygenic photosynthesis consists of light and dark reactions, the former of which starts from photon absorption and drives the photosynthetic electron transport chain originating from water up to the final electron carrier protein, ferredoxin[1] (Fd). Four large membrane protein complexes, Photosystem II[2], Cytochrome (Cyt) $b_6f$[3], Photosystem I[4] (PSI), and NADH-like complex I[5] (NDH-1) operate in the electron transport chain within the thylakoid membrane. Plastoquinone, plastocyanin (Pc), Cyt $c_6$, and ferredoxin (Fd) act as mobile electron carriers to shuttle electrons between these membrane complexes. These electron carriers form transient complexes with their redox partners. The electron transfer events should be sequential, occur with a high degree of specificity and in a kinetically efficient manner. In short, the rate of electron transfer is key to maximum performance of the complete electrochemical reaction. Intermolecular interaction of lipophilic organic molecules, plastoquinone or quinone analogs, between Photosystem II and Cyt $b_6f$ have been well characterized by kinetics[6] and X-ray crystallography[7]. However, protein–protein interactions of the water-soluble electron carriers Pc and Fd with Cyt $b_6f$, PSI, and NDH-1 are difficult to study due to the large molecular size of the redox-dependent structures.

PSI executes a light-driven fast charge separation for the transfer of electrons across the thylakoid membrane[8]. With a quantum efficiency of close to 100%, PSI is the most efficient energy converter found in nature[9]. Using excitation energy funneled to it by the surrounding antenna pigments in the PSI reaction center, charge separation takes place at a pair of chlorophyll $a$/chlorophyll $a'$ molecules referred to as P700[4]. The activated electron is then transferred through the electron transfer chain (ETC), and relayed via the [4Fe-4S] clusters $F_A$ and $F_B$ to the downstream water-soluble electron acceptor Fd at the stromal side[10]. The strong reductant Fd provides electrons to a variety of downstream reactions such as production of NADPH, nitrogen and sulfur assimilation, or fatty acid desaturation[11]. Oxidized P700 is subsequently reduced by a luminal electron donor protein, Cyt $c_6$ or Pc, launching the next round of electron transfer[12]. Cyanobacterial PSI is primarily a homotrimer, although monomeric[13] or tetrameric[14] forms are present. Each PSI protomer comprises up to 12 subunits that host more than 100 prosthetic groups, which make up a third of the total mass of the complex[15]. The X-ray structure of trimeric cyanobacterial PSI from *Thermosynechococcus elongatus* (formerly *Synechococcus elongatus*) was determined in 2001 at 2.5 Å resolution (PDB ID: 1JB0[4]), and in 2018 that from *Synechocystis* sp. PCC 6803 also at 2.5 Å resolution (PDB ID: 5OY0[16]). Later, several other PSI structures, including green plant-type PSI complexed with light-harvesting chlorophyll proteins I (LHCI), were revealed at a higher resolution by X-ray crystallography and also by cryogenic electron microscopy (cryo-EM); namely, PSI-LHCI from pea at 2.4 Å (PDB ID: 7DKZ[17]), PSI-LHCI-LHCII (PDB ID: 7D0J[18]) from green algae, and two PSI-LHCI complexes from a diatom at 2.38 and 2.4 Å resolution (PDB IDs: 6LY5[19], 6L4U[20]). After the resolution revolution[21] in cryo-EM, two more cryo-EM structures of cyanobacterial PSI at slightly higher resolution were published; namely trimeric PSI from *Halomicronema hongdechloris* C2206 at 2.35 Å (PDB ID: 6KMW[22]), and tetrameric PSI from a heterocyst-forming *Anabaena* sp. PCC7120 at 2.37 Å (PDB ID: 6K61[14]).

To better understand the PSI-related transmembrane electron transfer mechanism, various methods such as co-crystallization or chemical cross-linking were applied to PSI with its electron transfer partner(s). Fd:PSI was crystallized in 2002, despite being a non-covalent electron transfer complex[23]. In 2018 our group reported the first X-ray structure of an Fd:PSI complex at 4.2 Å resolution[24]. The structure confirmed Fd-binding sites on the stromal side of PSI, which were previously suggested by site-directed mutagenesis[25] and kinetic analysis[26]. In this Fd:PSI X-ray structure, gallium-substituted ferredoxin (Ga-Fd), whose protein structure is identical to that of native Fd[27], was used to fix the redox state of bound Fd to the oxidized form even when illuminated by light. Recently, several non-covalent electron transfer complex structures of PSI determined by cryo-EM have been reported, such as PSI:Fd (PDB ID: 7S3D[28]), PSI-IsiA complex with bound flavodoxin (PDB ID: 6KIF[29]), and the triple complex of Pc:PSI-LHCI:Fd (PDB ID: 6YEZ[30]). However, low local resolution of bound mobile carrier proteins prevented a detailed analysis at the residue level except for the Pc:PSI-LHCI supercomplex at 2.74 Å (PDB ID: 6ZOO[31]) for the binding mode of the luminal electron donor Pc. Despite extensive efforts[32] to visualize the electron transfer complex of PSI: Cyt $c_6$, thus far neither X-ray nor cryo-EM structure of the PSI:Cyt $c_6$ complex has been reported.

Higher resolution structures of PSI together with its electron donors and acceptors under a controlled redox state are required to better understand the protein–protein interactions involved in the electron relay system. Here, together with thermodynamic measurements on Fd binding to PSI, we describe the structure of cyanobacterial PSI from *Thermosynechococcus elongatus* BP-1 bound with its electron transfer partners Fd and Cyt $c_6$ as analyzed by single-particle cryo-EM at an overall resolution of 1.97 Å.

## Results

**Structure determination**. Purified PSI trimer sample was optimized for single-particle cryo-EM by detergent exchange to GDN[33] and removal of excess free detergent by GraDeR[34] yielding a highly homogeneous preparation at high concentration (Supplementary Fig. 1a, b). In the first step, parameters for cryo-grid preparation were optimized for high-resolution imaging of the PSI trimer alone. Optimal ice thickness and particle distribution were observed at a PSI trimer protein concentration of 30 mg/ml, whereas cryo-grid quality deteriorated at values below or above this concentration. In a second step, Ga-Fd was added at the previously for crystal growth optimized slightly basic pH of 8.0 and a molar ratio of 1.0:1.1 (PSI:Fd)[27], while insufficient yield for the purification of Cyt $c_6$ limited its molar ratio to 1.0:0.4 (PSI:Cyt $c_6$) resulting in the molar ratio of 1.0:1.1:0.4 (PSI:Fd:Cyt $c_6$) for the triple mixture used for final cryo-grid preparation. All cryo-grid screening was performed using a 200 kV cryo-TEM (Talos Arctica, TFS). Cryo-grids exhibiting large areas of suitable vitreous ice thickness with high particle density (Supplementary Fig. 2b) were then transferred to a 300 kV cryo-TEM equipped with a cold-FEG, an in column Ω energy filter and a K3 direct electron detector (CRYO ARM 300, JEOL; Gatan). Image processing in Relion 3.1[35] (Supplementary Fig. 2) yielded a final 3D density map calculated from 207,142 particles at an overall resolution of 1.97 Å, which was of sufficient quality to visualize side-chain densities and water molecules at the PSI:Fd binding interface. Supplementary Table 1 describes cryo-EM data collection, refinement, and validation statistics. Both post-processed maps with C3 and C1 symmetry were deposited to the Electron Microscopy Data Bank (EMDB[36], accession number: EMD-31605). Raw images were uploaded to the Electron Microscopy Public Image Archive (EMPIAR[37], accession number: EMPIAR-10928).

**Overall architecture of the complex**. Viewed along the membrane plane, the Fd-bound PSI trimer resembles the shape of a clover leaf with a diameter of ~200 Å and a total mass of 1110 kDa (Fig. 1). Because no significant differences were

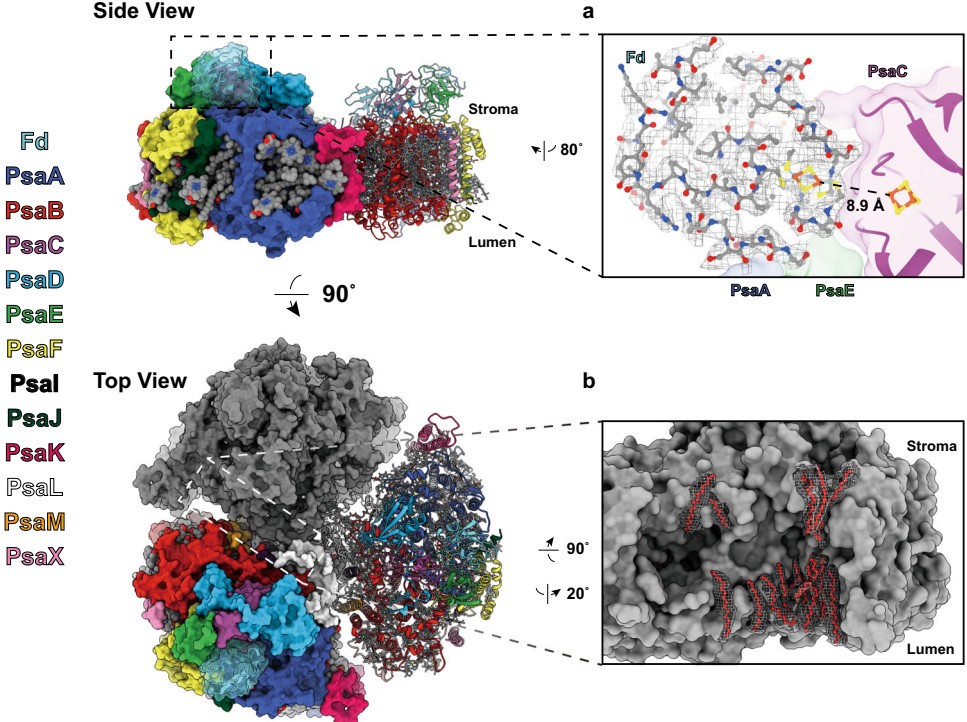

**Fig. 1 Overall architecture of the PSI:Ga-Fd complex viewed parallel to the thylakoid membrane ("side-view") and from the stromal side ("top-view").**
The Fd-bound PSI trimer complex model in C3 symmetry shown with each of the three protomers either in gray surface, or a surface colored by the subunit chains (PsaA-blue, PsaB-red, PsaC-purple, PsaD-light blue, PsaE-green, PsaF-yellow, PsaI-dark purple, PsaJ-dark green, PsaK-pinkish red, PsaL-white, PsaM-orange, PsaX-pink, Fd-cyan) or colored in a cartoon style. Ligands are presented in sphere and stick style separately, colored according to the element. **a** Close-up view of the ferredoxin (Fd) binding interface and the corresponding density map for Fd. **b** Modeled lipid acyl chains between PSI protomers and their corresponding density map.

identified between the non-symmetrized (C1) density map at 2.07 Å and the symmetrized (C3) map at 1.97 Å resolution, the latter map was used for atomic model building. The crystal structure of PSI solved to 2.5 Å resolution (PDB ID 1JB0[4]) and that of Fd solved to 1.5 Å resolution (PDB ID 5AUI[27]) from *T. elongates* were used as reference models. Our model (PDB ID: 7FIX) of the PSI:Fd complex was refined against the final density map determined at 1.97 Å with strict three-fold symmetry (C3) (Supplementary Fig. 3). In our model, all 12 subunits of each *T. elongatus* PSI protomer (PsaA, PsaB, PsaC, PsaD, PsaE, PsaF, PsaI, PsaJ, PsaK, PsaL, PsaM, and PsaX) were included in addition to three Fds equally bound at the stromal side of the PSI trimer to PsaA, PsaC, and PsaE. A close-up view (Fig. 1a) shows the atomic model of PSI-bound Fd, which fits well with the corresponding density map. The [2Ga-2S] cluster of Fd and the [4Fe-4S] cluster of $F_B$ in PsaC subunit, drawn in ball and stick style, were positioned at an edge-to-edge distance of 8.9 Å.

Supplementary Fig. 4a shows the local resolution of the density map in both top- and side-view. For each PSI protomer on the luminal side of the PSI trimer at the membrane facing side of PsaB and in the vicinity of the N-terminal region of PsaF a globular density was observed at a map threshold similar to that of the PSI surrounding detergent belt. The positions and features of all three densities are identical, they were already present in the initial de novo model (Supplementary Fig. 2e), and most likely stem from loosely bound Cyt $c_6$. These three densities did not improve in local resolution or definition when subjected to masked 3D classification and refinement. Rigid-body modeling of the cyanobacterial Cyt $c_6$ structure using Phenix[38] showed the density to be of a sufficient size for Cyt $c_6$ (Supplementary Fig. 4b, c). Nonetheless, ambiguity of its orientation prompted us not to

model the ternary complex of Fd:PSI:Cyt $c_6$ in our final atomic coordinates (7FIX) shown in Fig. 1.

The complete model contains a total of 39 polypeptide chains, 285 chlorophyll *a*, three chlorophyll *a′*, 72 β-carotenes, six phylloquinones, nine [4Fe-4S] iron–sulfur clusters, three $Ca^{2+}$ ions, nine phosphatidylglycerols, three digalactosyldiacylglycerols, 348 water molecules and 51 lipids built without a headgroup for PSI, and three polypeptides and three [2Ga-2S] clusters for Fd. Notable differences to our reference models included identification of the previously monogalactosyldiacylglycerol modeled Lipid II as digalactosyldiacylglycerol, the extension of Lipid IV acyl chains and a change of PsaL residue 143 from a leucine to a serine, which now matches the primary sequence data bank entry for PsaL (UniProtKB - Q8DGB4). Our map allowed us to extend the terminal regions by 39 amino acid residues, namely Lys11–Val12, Gly263–Ile265 in PsaA, Gly740 in PsaB, Thr5–Val19, Ala33, Pro44–Phe54, Gln78–Leu83 in PsaK, and Glu3 in PsaL. We did not observe any density for a chlorophyll bound to PsaM (CLA 1601 in 1JB0), which was not included in our model. Furthermore, two densities in PsaK and PsaA were partially modeled as the newly identified β-carotenes BCR102 of PsaK and BCR855 of PsaA. However, since both densities do not show clear polyene chain methyl bumps, we cannot rule out that these densities originate from other similar molecular species such as lipids. Moreover, a new chlorophyll was identified in PsaJ and included in our model as PsaJ CLA1307. Finally, a total of 17 new lipid acyl chains for each monomer–monomer interface could be built into our model (Fig. 1b). All additions and modifications are presented in Supplementary Fig. 5 together with representative maps and models (Supplementary Fig. 6).

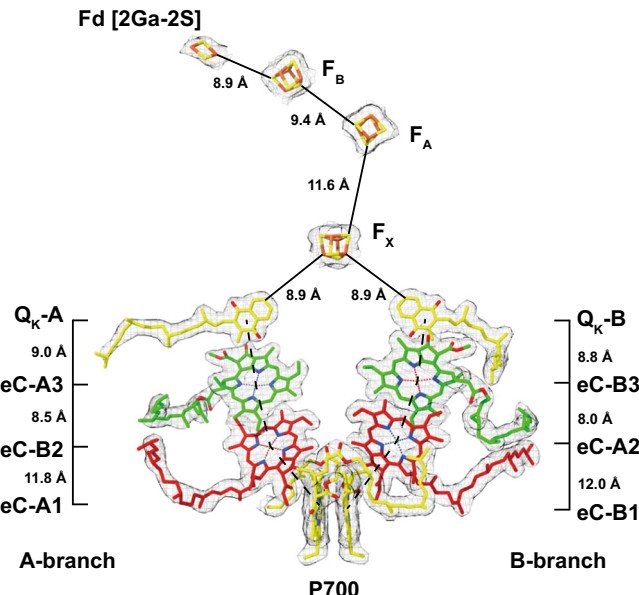

**Fig. 2 Core Components of the electron transfer chain (ETC) of the PSI:Fd complex with the corresponding cryo-EM densities.** Distances between cofactors were measured 'center-to-center' from magnesium atoms of neighboring chlorophylls to the center of the phylloquinone carbonyl rings for the A-branch and the B-branch (dashed lines) or 'edge-to-edge' between phylloquinones and iron–sulfur clusters (solid lines).

**The electron transfer chain components.** All components of the ETC including [2Ga-2S] of Fd as well as their density maps excluding the heme of Cyt $c_6$, are presented in Fig. 2 with their intermolecular distances calculated either as center-to-center (P700 to phylloquinone) or edge-to-edge (phylloquinone to Fd [2Ga-2S]). Branch A and B of the ETC are both composed of two closely stacked chlorophyll $a$ and one phylloquinone molecules located equidistant (8.9 Å) to the [4Fe-4S] cluster $F_X$, which is coordinated by both PsaA and PsaB. PsaC hosts the [4Fe-4S] clusters $F_A$ and $F_B$, the two last components of the ETC, of which $F_B$ is the final electron acceptor and the point of electron transfer to the docked soluble electron acceptors, Fd or Flavodoxin (Fld). The high local resolution of our PSI:Fd density map enabled us to measure co-factor distances in the ETC with a good degree of confidence. The observed edge-to-edge distances between phylloquinones and the three [4Fe-4S] clusters are in good agreement with those previously determined in the 2.5 Å X-ray crystal structure (PDB ID: 1JB0). The distance of 8.9 Å between $F_B$ and [2Ga-2S] of Fd is short enough for efficient electron tunneling and matches the distances that were found in the PSI-IsiA-Fld supercomplex[29]. Moreover, this distance is also equivalent to the mean of the three differing distances observed in our previous PSI:Fd X-ray crystal structure[24].

**Intermolecular interactions between PSI and Fd.** In our structure, three Fd molecules are tightly bound to the stromal side of the PSI trimer. Even in the map contoured at a density level close to noise no further density indicative of additional Fd molecules bound at other stromal sites of PSI can be discerned. Each Fd molecule binds to one PSI protomer in an identical fashion making close contact to subunits PsaA, PsaC, and PsaE, whereas no direct interaction with PsaD or PsaF could be found. The local resolution of ~2.5 Å and well-defined map features at the PSI:Fd interface allowed us to model amino acid side chains and several water molecules with confidence. Amino acid residues and water molecules involved in direct interaction at the PSI:Fd interface

were defined using the DIMPLOT program of the LigPlot+[39, 40] suite. Based on DIMPLOT analysis, a total of 18 residues within the PSI subunits (Arg36, Arg40, Thr60 of PsaA; Ile11, Gly12, Gln15, Arg18, Lys34, Ala35, Ala56, Pro58 of PsaC; Arg3, Ile37, Arg39, Ser53, Asn56, Thr57, Asn59 of PsaE) and 20 residues within Fd (Ser60, Asp61, Asp67, Ile70 of PsaA; Phe38, Ser39, Cys40, Ala44, Cys45, Thr47, Phe64, Tyr97 of PsaC; Glu23, Tyr24, Asp27, Glu31, Cys40, Arg41, Ala42, Ser63 of PsaE) were identified as being involved in binding interactions (Fig. 3 and Supplementary Fig. 7). Four residues (Glu23, Asp27, Phe38, Thr47) in Fd were newly identified as being involved in the binding interaction, whereas four residues of Fd previously assigned to be interacting with PSI (Gln62, Asp66, Glu71, Tyr81)[24] were not found to do so in this study. A total of six bridging water molecules (HOH202 to Fd:PsaA, HOH203, HOH204, HOH205 to Fd:PsaC, HOH101, HOH201 to Fd:PsaE) were identified close to the binding surface, offering potential hydrogen bond linkages between PSI and Fd. A comparison of results from X-ray crystallography, NMR transferred cross saturation, and this study on the question which residues of Fd are interacting with PSI upon binding is presented in Fig. 4.

Two important PSI:Fd binding interfaces are shown in Fig. 5. A pair of cation-π interactions between Phe38 of Fd with Lys34 of PsaC and Arg41 of Fd were identified (Fig. 5a). Arg39 of PsaE, which interacts with Arg41 of Fd through hydrophobic interactions, makes additional interactions with Asp27 and Tyr24 of Fd. Residue Gln15 of PsaC forms the greatest number intermolecular interactions between PSI and Fd, and may be strategically situated within the vicinity of electron donating and accepting clusters (Fig. 5b). Gln15 of PsaC is engaged in hydrophobic interactions with Cys45 and Phe64 of Fd and also makes a hydrogen bond to Ala44. In addition, *via* a hydrogen bond to HOH205, Gln15 of PsaC is connected to the C-terminal Tyr97 of Fd. Tyr97 in turn makes a cation-π interaction to Arg18 of PsaC. Tyr97 of Fd exhibits a well-defined density, which is surprising given the known flexibility of the C-terminus (Supplementary Fig. 8), indicating strong interactions between Fd Tyr97 and PsaC.

**Structural and thermodynamic changes upon PSI:Fd complex formation.** To visualize dynamic structural changes upon complex formation between PSI and Fd, we used python script "modevectors" running on the PyMOL Molecular Graphics System (Version 2.3.2, Schrödinger, LLC). In Fig. 6, atomic models of PSI before (PDB ID: 1JB0) and after Fd-binding (current study, PDB ID: 7FIX) were superposed based on α-carbon positions of the stable PsaA/PsaB heterodimer. Displacements of α-carbons >0.7 Å before and after complex formation are highlighted by arrowheads for one protomer. Because individual side chains may be involved in the crystal packing for X-ray structures of PSI or Fd, we intentionally used the α-carbon positions of main chain structures for comparisons and the Cα-baton models without side chains were shown in Fig. 6. The analysis shows that perturbations caused by Fd binding and the presence of Cyt $c_6$ are not limited to the stromal side, but also propagate to the luminal side. When comparing the atomic model of Fd in our complex structure (PDB ID: 7FIX) against the atomic model of free Fd (PDB ID: 5AUI)[27] by displaying displacements of >0.7 Å at the α-carbon (Fig. 6c), we found that the cluster proximal region remained unperturbed whereas three distal regions (in dashed circles) underwent structural changes after binding to PSI.

Isothermal titration calorimetry (ITC) measurements were performed to obtain thermodynamic parameters for the interaction between Ga-Fd and the PSI trimer. To avoid potential interference of free detergent micelles with the ITC measurements[41], the PSI trimer samples were prepared using the GraDeR[34] approach as it was done

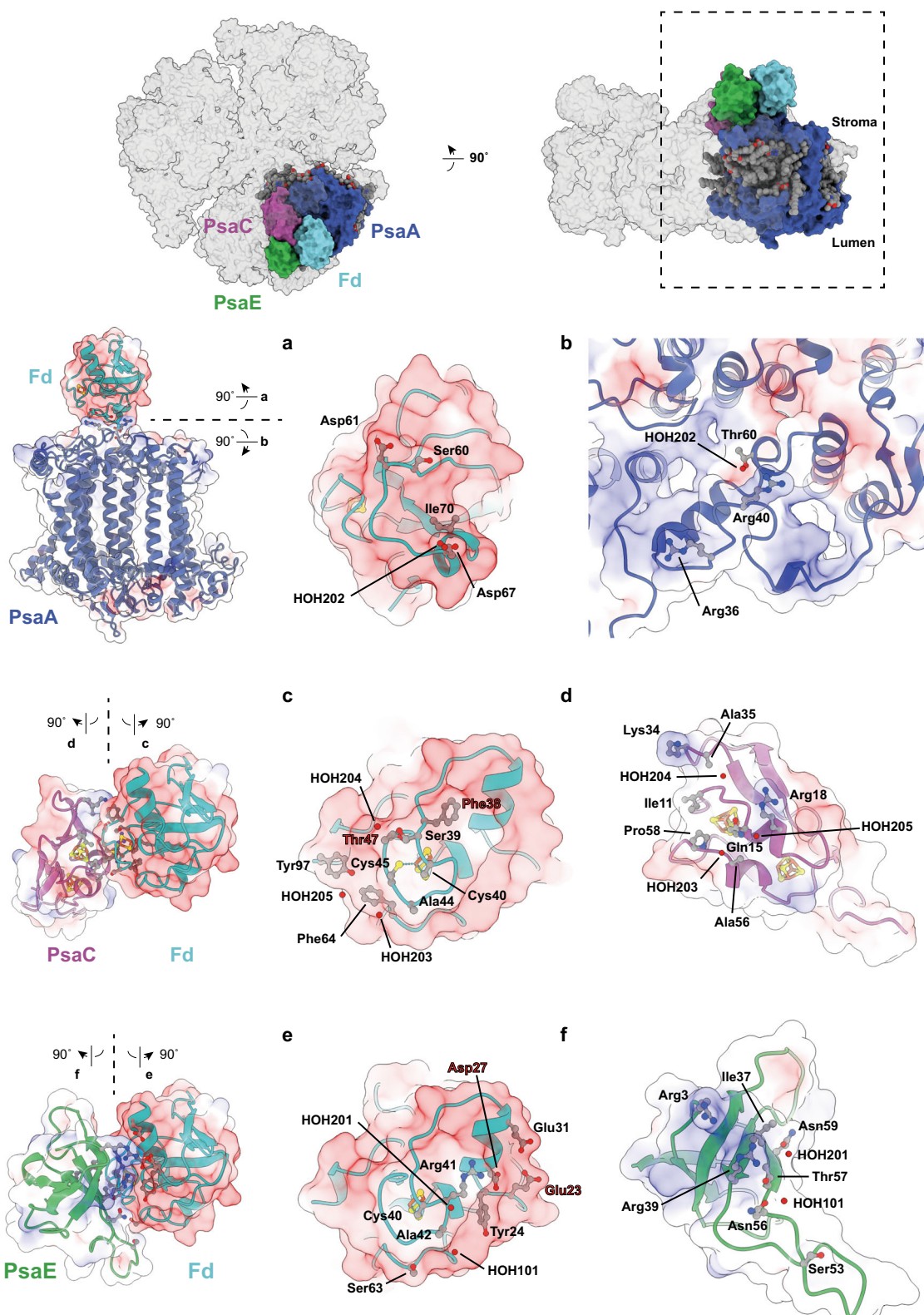

**Fig. 3 Ferredoxin (Fd) and PSI binding interfaces.** Each interface involved in binding is presented separately in open-book style. Directly interacting amino acid residues and water molecules are labeled. **a**, **b** For Fd:PsaA, **c**, **d** for Fd:PsaC, **e**, **f** for Fd:PsaE. Newly identified residues involved in the interaction are labeled in red. All protein surfaces are colored according to the electrostatic potential (positive in blue, negative in red).

for single-particle cryo-EM. Representative results of the ITC thermograms, the binding isotherm of interactions between Ga-Fd and PSI trimer, and thermodynamic parameters are shown in Fig. 7 and Table 1. The ITC analyses demonstrate that the binding stoichiometry value is ~3 (2.9 ± 0.2), indicating Fd is bound to each PSI protomer at a ratio of 1:1; which is in agreement with our cryo-EM density map. The $K_d$ value was in the sub-micromolar order (758.0 ± 123.5 nM), suggesting that the inter-protein affinity between

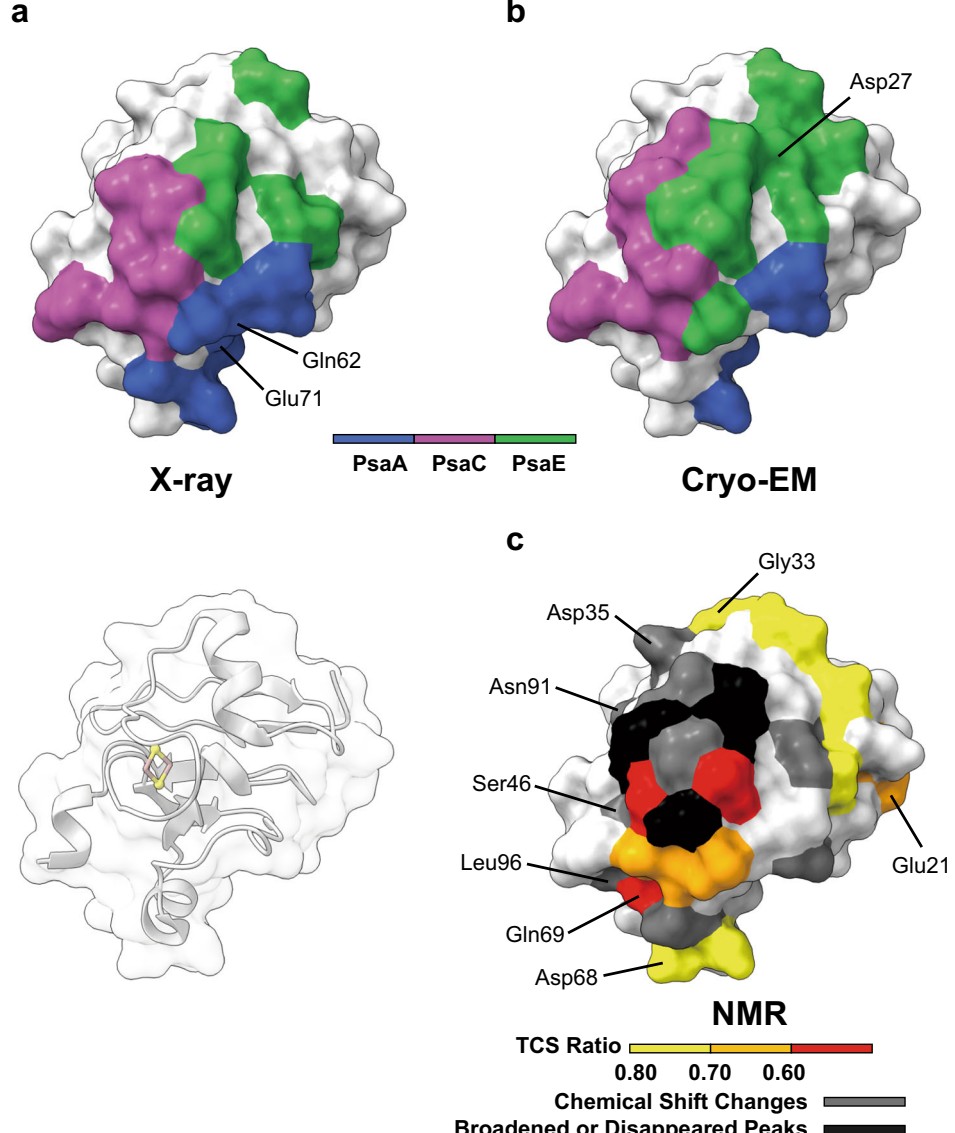

**Fig. 4 The PSI:Fd interactive residues in *T. elongatus* Fd.** The interactive residues were determined by **a** Co-crystallized PSI:Fd X-ray diffraction (PDB ID: 5ZFO[24]), **b** Cryo-EM (this study), and **c** NMR transferred cross-saturation. Unique residues determined by each method are highlighted by the corresponding labels (residues at the dorsal side were omitted).

Ga-Fd and PSI is strong. A series of titration of Ga-Fd to PSI trimer generated positive ITC peaks followed by gradual saturation (Fig. 7a), indicating the endothermic intermolecular interactions between Ga-Fd and PSI. Overall, the Fd-PSI interactions showed a thermodynamically unfavorable positive $\Delta H$ (1.1 ± 0.1 kcal/mol) value and a stronger favorable positive $T\Delta S$ (9.5 ± 0.1 kcal/mol) value, demonstrating complex formation is entropy-driven.

## Discussion

Our cryo-EM structure of the PSI:Fd complex is based on a density map with a strict three-fold rotational symmetry at an overall resolution of 1.97 Å. The improved resolution might be the result of four factors: (i) exceptional purity and homogeneity of our PSI preparation at high concentration, (ii) a strictly controlled redox state of the transient PSI:Fd electron transfer complex and the presence of Cyt $c_6$ (iii) optimized membrane protein complex preparation using the GraDeR method, and (iv) use of an advanced cryo-electron microscope CRYO ARM 300 equipped with a cold-field emission gun, an in-column Ω energy

filter and a K3 direct detection camera (JEOL, GATAN). One noteworthy finding from our high-resolution cryo-EM map is the visualization of several stick-shaped densities between each PSI protomer (Fig. 1b and Supplementary Fig. 5), indicating stable and specific packing of lipid tails. Although thylakoid membranes contain various lipids, including PG, MGDG, DGDG, and SQDG[42], the original 2.5 Å X-ray structure[4] only described three phospholipids and one galactolipid internally bound to the major subunits PsaA and PsaB. In all, seventeen newly identified fatty acid tails were found in our structure despite excluding ambiguous densities that could not be clearly distinguished from chlorophyll phytol chains or were too short or not perpendicular to the membrane plane. These newly found inter-protomer lipids were not assigned to specific types due to their missing headgroup densities, indicating their mobility. Visible fatty acid tails were tightly attached to hydrophobic amino acid residues and antenna ligands (chlorophylls and carotenoids) implying their potential function in stabilizing the trimerization of PSI. These findings explain discrepancies between the number of associated lipids

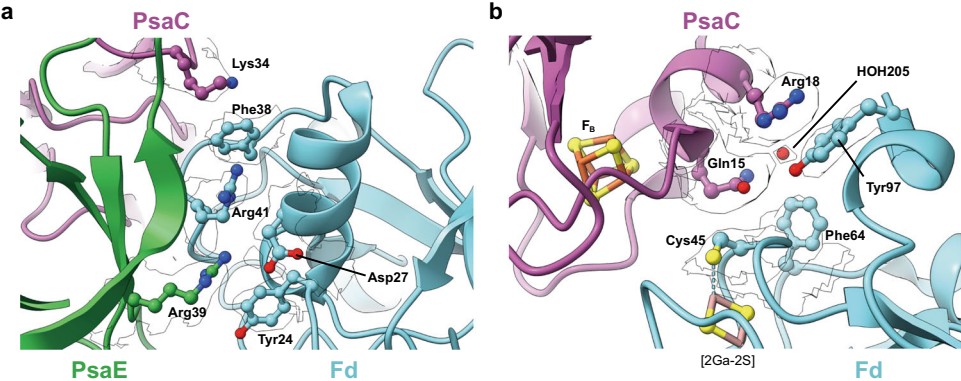

**Fig. 5 Key sites of Fd-PSI interaction. a** Nexus of interactions that contribute to the binding affinity of PSI with Fd. Crucial PSI residues Lys34 of PsaC and Arg39 of PsaE and their respective Fd-binding partners are highlighted. PsaC in purple ribbon, PsaE in green ribbon, Fd in cyan ribbon and their interacting residue side chain atoms in ball and stick with their corresponding density in silhouette. **b** Hub of interactions around Gln15 of PsaC and with Cys45, Phe64, and Tyr97 of Fd. These residues undergo a conformational change upon Fd reduction. PsaC in purple ribbon, Fd in cyan ribbon and interacting residues side chain atoms in ball and stick with their corresponding density in silhouette. Interfacial water 205 is shown as a red ball. Iron–sulfur cluster FB of PSI and the gallium-sulfur cluster of Fd is depicted in ball and stick format.

determined by structural and biochemical analyses[43]. Indeed, our findings suggest an additional function of lipids in structurally tightening the protomer interface to enable stable formation of oligomeric PSI and to accommodate protruding antenna ligands especially at the inter-protomer region. To date, these aspects have thus far only been considered in detail for the structure of PSII[2, 44].

Herein, we describe the high-resolution cryo-EM structure of cyanobacterial PSI in the presence of oxidized Fd and reduced Cyt $c_6$. The sample for structural determination was prepared by mixing the three proteins without cross-linking and only by a brief incubation in the dark before cryo-grid preparation. Furthermore, we used gallium-substituted Fd (Ga-Fd), which is structurally identical to the wild-type oxidized Fd but cannot be reduced[27, 45]. The resulting uniform redox state at the acceptor side of PSI eliminates the possibility of unexpected redox reactions between Fd and PSI triggered by light absorption. Hence, the redox state of the complex was that of PSI just before electron transfer from PSI to Fd. In this study, we could visualize weak, however, for all three PSI protomers identically positioned extra densities in the vicinity of the N-terminal region of PsaF assigned to Cyt $c_6$ (Supplementary Fig. 4b, c) which were already clearly present in the de novo 3D initial model of our cryo-EM structure (Supplementary Fig. 2e). The very low local resolution could be a consequence of the low stoichiometry of 1.0:0.4 (PSI:Cyt $c_6$) in the mixture used for cryo-grid preparation. However, the previously attempted unsuccessful visualization for Cyt $c_6$ bound to the *T. elongatus* PSI trimer at the much higher PSI:Cyt $c_6$ stoichiometry of 1:10 under reducing buffer conditions and in the presence of cross-linker[32] suggests that stoichiometry and redox state of the sample buffer alone might be not sufficient to explain the only close to noise level of the here detected extra luminal density.

Surprisingly, Cyt $c_6$ was not found close to the primary electron acceptor of P700 in PSI (center-to-center distance P700 to Cyt $c_6$ ~55 Å) but at a distance sufficient to prevent productive electron transfer. Although the lack of local resolution prevented determination of the orientation of Cyt $c_6$, we believe the site itself is a physiologically relevant secondary binding site for Cyt $c_6$ that is in clear variance from that of plastocyanin visualized in the recently reported PSI:Pc structure (PDB ID: 6ZOO[31]). In the structure of PSI:Pc, Pc was found to bind to the hydrophobic surface created by PsaA and PsaB subunits, and the N-terminal extension of PsaF specific for the vacuolar plant placed Pc near P700 close enough

for productive electron transfer. Because cyanobacteria do not contain an N-terminal extension of the PsaF subunit for productive Pc/Cyt $c_6$ binding[46], it seems unlikely for cyanobacterial Cyt $c_6$ to bind to a position similar to that of higher plants. Previous attempts to visualize PSI with bound Cyt $c_6$ by cross-linking were unsuccessful[32] and to our knowledge the cryo-EM densities of Cyt $c_6$ in our study are the first experimental visualization of Cyt $c_6$ bound to PSI at this peripheral non-productive site.

What could be the physiological function of this newly discovered Cyt $c_6$ binding site? The Cyt $c_6$ binding site is clearly different from the productive Pc/Cyt $c_6$ binding site. Results of electron transfer kinetic analysis from green algae and higher plants indicated biphasic[46, 47] features: a fast electron transfer phase relevant to the reduction of P700; and a slow phase responsible for the diffusion time of Pc/Cyt $c_6$ in the luminal space to the PSI reaction centers. The structural basis of this biphasic kinetics was assigned to the Lys-rich N-terminal extension of PsaF of green algae[46]. Recently, a cryo-EM structure of PSI from *Chlamydomonas reinhardtii* showed an extra density on the luminal side of the complex that suggests non-productive binding of Pc[48]. This additional region offers electrostatic interaction sites to electron donor proteins, which plays a key role in guiding them towards the P700 reaction center. The heterogeneous cross-linking result and kinetic features indicated the existence of both productive and non-productive binding sites[49]. The cross-linking test[50] performed in *Chlamydomonas reinhardtii* gave a similar result. Specifically, only 33% of Cyt $c_6$ performed fast electron transfer to P700, whereas the other 67% inactive cross-linking indicated the existence of a potential extra binding site. Such non-productive binding sites may be too distant from P700 for electron transfer, but nonetheless, guarantee a continuous fast supply of electron donors. A possible function of the Cyt $c_6$ binding site visualized here might be a non-productive binding site, analogous to the one found in vascular plants.

The existence of the two-phase reaction in cyanobacteria was disputed given that cyanobacterial PsaF does not contain the N-terminal extension found in PsaF of green algae and higher plants[12]. Recently, however, an in vivo P700 reduction kinetic study on the PSI from *Synechocystis sp.* PCC6803 showed similar kinetic features to those of green algae and higher plants[51]. The two-phase reaction could be the consequence of non-productive binding of the electron donors (Pc and Cyt $c_6$) on the luminal side of cyanobacterial PSI. The lack of extensions in cyanobacterial

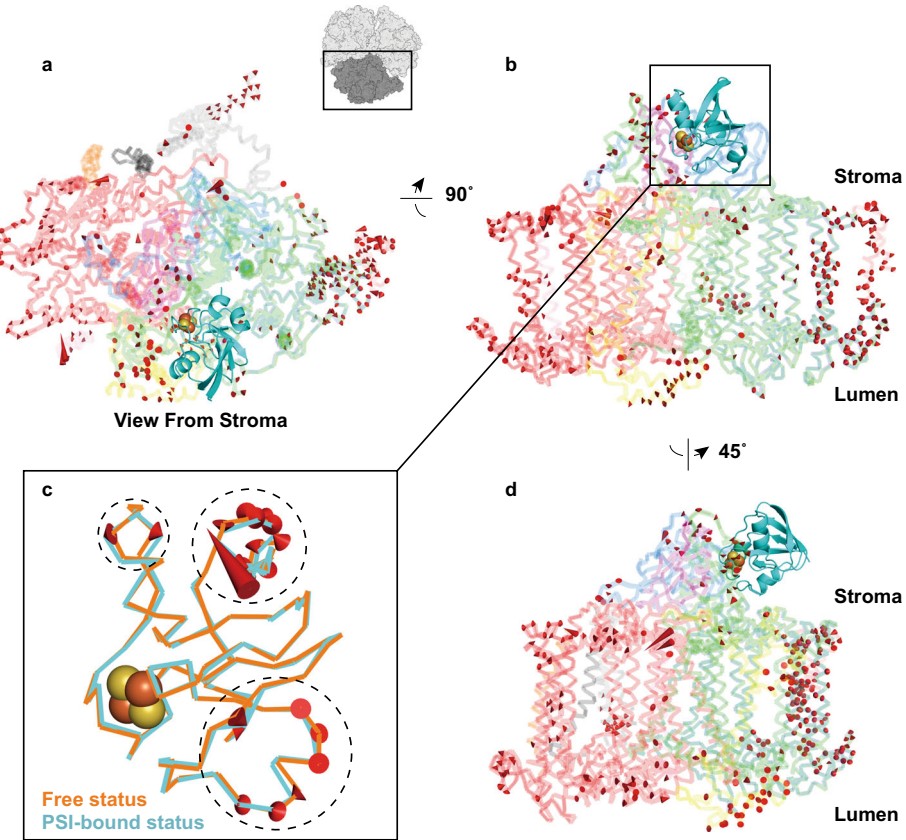

**Fig. 6 Structural changes induced by Fd binding. a, b, d** The Fd binding induced structural displacements in one PSI protomer from three different viewpoints. Atomic model of Fd-free PSI (PDB ID: 1JB0[4]) was superposed with our model of PSI:Fd (PDB ID: 7FIX) based on the Cα backbone coordinates of the PsaA/PsaB heterodimer. Red arrows represent displacements beyond 0.7 Å. **c** Structural displacements of Fds at free and PSI-bound status. Atomic model of unbound Fd (PDB ID: 5AUI[27], colored in orange) was superposed with PSI-bound Fd (colored in cyan). Red arrows highlight main chain displacements beyond 0.7 Å.

PsaF may make the Cyt $c_6$-binding specific but flexible, which could provide further explanation for the limited resolution of Cyt $c_6$ in our density map. This proposal is consistent with the low local resolution of bound Pc at the non-productive site of *Chlamydomonas* PSI[48]. It might also rationalize why in the previous cryo-EM study by Kölsch et al.[32] on *T. elongatus* PSI, cross-linking of PSI and Cyt $c_6$ combined with masked 3D refinement did not locate potential Cyt $c_6$-like densities at the related region on the PSI luminal side despite the much higher stoichiometry of Cyt $c_6$ to PSI (10:1) employed. Therefore, we surmise that one obvious difference between the study by Kölsch et al.[32] and ours is the addition of Ga-Fd. This suggests that the binding status of the electron acceptor (in our case, Fd) on the stromal side of PSI may be crucial for the binding of the electron donor protein (in our case, Cyt $c_6$) on the luminal side and as a consequence the ability to visualize it. Thus, the electron transfer status on the stromal side of PSI could affect the binding mode of electron donors on the luminal side *via* conformational-alteration-based transmembrane signaling. Future studies that specifically focus on optimizing specific binding of Cyt $c_6$ at high occupancy will hopefully clarify these aspects of the cyanobacterial PSI-mediated electron transfer.

The binding of Fd to PSI has been studied in detail by mutagenesis of the stromal subunits. These studies identified key residues involved in the interactions that allow fast inter-protein electron transfer (summarized in Supplementary Table 2). Our atomic model provides the structural basis for the important role played by these key amino acid residues and can explain their mutational effects. At the Fd-binding interface on PsaE, the

previously established key interactions of Arg39[52], whose mutation to a glutamine completely disrupts Fd binding, is now rationalized by its electrostatic interaction with Asp27 and hydrophobic interactions with Tyr24 and Arg41 of Fd (Fig. 5a and Supplementary Fig. 7c). At the interface with PsaC, Lys34 is engaged in hydrophobic interactions with Phe38 of Fd, which explains the inability of the PsaC K34D mutant[25, 53] to form the crucial cation-π interaction, completely disrupting fast electron transfer (Fig. 5a). Moreover, at the Fd-binding interface of PsaC near the iron–sulfur cluster $F_B$, Gln15 shows extensive direct interactions with Ala44, Cy45, Phe64, and also indirectly with Tyr97 through water molecule HOH205 (Fig. 5b), which likewise has also been identified as one of two key interaction patches for efficient binding of Fd[54] (Fig. 5).

An important question in the PSI-mediated electron transfer is how the bound Fd achieves fast dissociation from its PSI binding site after receiving an electron. The high-resolution X-ray crystallographic structure of *Anabaena* Fd in its oxidized and reduced state[55] suggested that the main structural changes upon reduction entails a flip in the peptide bond between Cys45 and Ser46 (*T. elongatus* numbering), a shift in the side chain of Phe64 and side chain movement of the C-terminal Tyr97. The nexus of interactions between PSI and Fd visualized in our structure centered around Gln15 of PsaC are disrupted by these reduction-induced conformational changes (Fig. 5b). The peptide plane flipping by itself would break the interaction between Fd-Cys45 and PsaC-Gln15 and the resulting shift of Phe64 might also disturb the interaction with Gln15. Finally, movement of the C-terminal Tyr97 could dislodge water-mediated hydrogen bonds associated

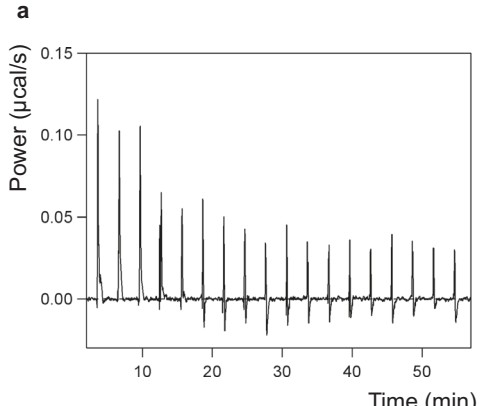

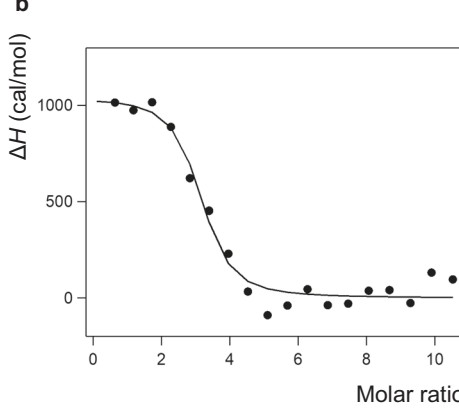

**Fig. 7 Isothermal titration calorimetry of Fd binding to PSI. a** Representative results of ITC thermograms, **b** The binding isotherm fitted using the one set of sites.

| | 1st measurement | 2nd measurement | 3rd measurement | Mean ± S.E.M. |
|---|---|---|---|---|
| **Table 1 Thermodynamic parameters of the interaction between Ga-substituted Fd and PSI trimer.** | | | | |
| $K_d$ (nM) | 703 | 994 | 577 | 758.0 ± 123.5 |
| $\Delta G$ (kcal/mol) | −8.4 | −8.2 | −8.5 | −8.4 ± 0.1 |
| $\Delta H$ (kcal/mol) | 1.0 | 1.1 | 1.2 | 1.1 ± 0.1 |
| $-T\Delta S$ (kcal/mol) | −9.4 | −9.3 | −9.7 | −9.5 ± 0.1 |
| $n$ | 2.9 | 3.2 | 2.5 | 2.9 ± 0.2 |

Three independent ITC experiments were performed at 25 °C. Each set of the thermodynamic parameters obtained by fitting normalized $\Delta H$ in the binding isotherm are shown without fit errors. Mean and S.E.M. values were also presented, indicating the averaged value and the standard error of the mean, respectively.
$K_d$ dissociation constant, $\Delta G$ change in Gibbs free energy, $\Delta H$ change in enthalpy energy, $\Delta S$ change in entropy, $n$ binding stoichiometry.

with Gln15. The enthalpic energy cost of losing this hub of interactions formed by Gln15 might then be sufficient to shift the energetic balance towards dissociation. Thus, our structure of the PSI:Fd interface in close vicinity to iron–sulfur clusters engaged in inter-protein electron transfer provides a convincing scenario to explain the mechanism of electron transfer-induced dissociation of Fd from PSI.

In this study, we used the crystal structures of wild-type PSI (PDB ID: 1JB0[4]) and wild-type Fd from *T. elongatus* BP-1 (PDB ID: 5AUI[27]) as a starting reference for modeling and refinement. Although some differences originating from the significant improvement of resolution were found in comparison to our previous analysis[24] by X-ray crystallography at 4.2 Å resolution and NMR (PDB ID: 5FZ0, BMRB:11596, Fig. 1), the observed differences in interacting residues between Fd and PSI are minor and in agreement with our earlier interpretation. Here, we were able to include side chain information into the assignment of interactions of Fd with PSI for the first time. In our previous X-ray structure of the PSI:Fd complex we concluded that PsaD does not interact with Fd, but this observation conflicted with the results of site-directed mutagenesis in *Synechocystis* PSI[56]. However, our updated high-resolution cryo-EM structure now strongly corroborates the lack of interaction between PsaD and tightly bound Fd. Based on our X-ray crystal structure of the PSI:Fd complex, we previously proposed that cyanobacterial PsaF can act as a transmembrane signal transducer activated by Fd-binding[24]. While the observed structural changes are only small, the absence of lattice contacts with crystallographic neighbors in this study, which also visualized structural changes in PSI upon Fd binding and in the presence of Cyt $c_6$ on both the stromal and luminal side (Fig. 6), supports our previous X-ray crystal structure-based proposal that small transmembrane structural changes might signal the binding of Fd to the luminal side.

Though the physiological relevance of the structural changes detected here cannot be ascertained, the idea of a transmembrane relay system for the coordination of electron transport through PSI remains attractive.

Surprisingly, six water molecules mediating hydrogen bonds between Fd and PSI can be visualized in the structure (Fig. 3 and Supplementary Fig. 7). Before complex formation both Fd and the stromal subunits of PSI must be fully hydrated by solvent. After forming the specific electron transfer complex between Fd and PSI, water molecules from the hydration shells except for the remaining six water molecules must be excluded from the interface. To obtain thermodynamic insights into complex formation of Fd with PSI including hydrating water molecules, ITC measurements were performed. Based on analysis of the thermodynamic parameters, the formation of the Fd:PSI complex is akin to that of the Fd:FNR[57] and the Fd:GmHO[58] type ($\Delta H > 0$ and $\Delta S > 0$) but not to that of the Fd:SiR type[59] ($\Delta H < 0$ and $\Delta S > 0$). Binding of Fd to its partner protein (in this case, PSI) appears to be entirely driven by entropy. Attractive electrostatic interactions, polar interactions, and Van der Waals' force-driven interactions might contribute to a negative $\Delta H$ ($\Delta H < 0$), but detachment of water molecules bound to interfaces for complexation of Fd and PSI will cost thermodynamically to give a positive value for the net $\Delta H$ ($\Delta H > 0$), as also observed in the Fd:FNR complex formation[57, 60]. Nonetheless, conformational disorder accompanying the displacement of water molecules from the interface can make complex formation energetically favorable due to the associated increase in entropy. The ITC analysis demonstrates a strong inter-protein affinity between Ga-Fd and PSI with a $K_d$ value of ~0.8 μM, which closely corresponds to the affinity of native Fd to PSI as measured by flash-absorption spectroscopy[24]. Last, the binding stoichiometry ($n$ value) calculated based on ITC measurement results was approximately three,

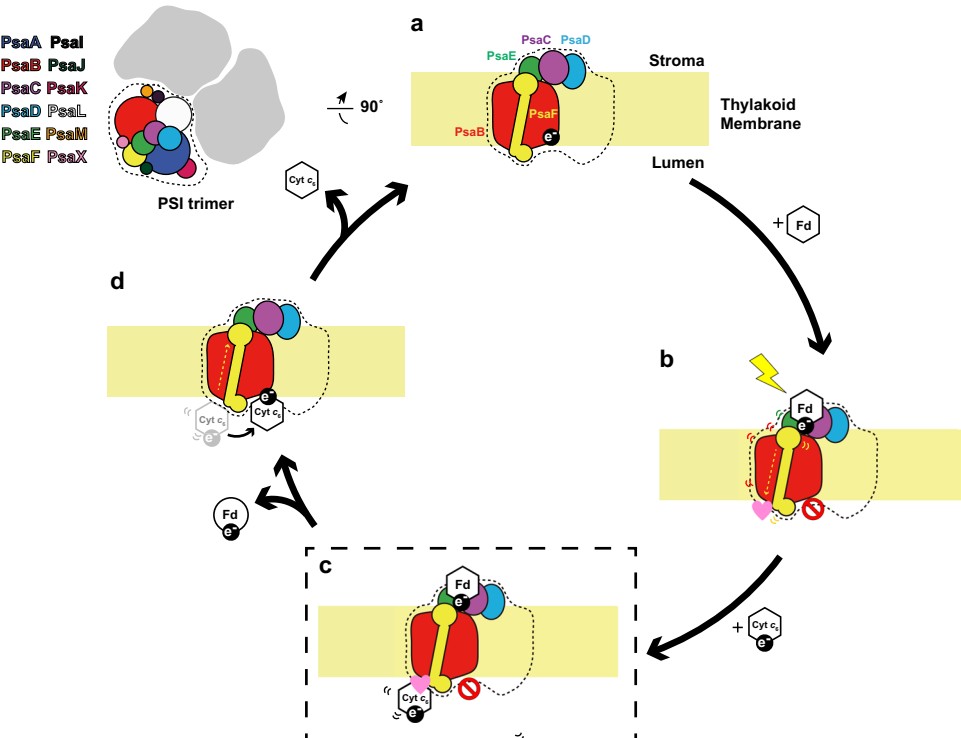

**Fig. 8 Schematic diagram of the proposed PSI electron pumping cycle. a** Reduced Cyt $c_6$ and oxidized Fd are in a free unbound state. **b** Light-induced charge separation across the membrane causes oxidized Fd to bind on the stromal side and opens up non-productive docking sites for Cyt $c_6$ on the luminal side. The pink heart-shaped symbol represents an activated binding site and the 'closed' traffic sign represents a closed reaction center. **c** Cyt $c_6$ is bound to luminal non-productive docking sites while permanently oxidized Ga-Fd remains bound at the stromal side with the dashed line frame indicating the structure determined in this study. **d** The excited electron has left PSI with its soluble carrier Fd and the reduced Cyt $c_6$ has moved to the productive P700* proximal binding site for electron transfer and subsequent detachment from PSI.

meaning that one Ga-Fd binds to one PSI protomer in solution, which is consistent with the results of our cryo-EM structure.

Based on kinetic analyses and our updated structural study, we propose a mechanism of electron transfer complex formation involving five steps as follows (also refer to the cartoon in Fig. 8). (i) Fd and Cyt $c_6$ are initially in a free unbound state before electron transfer (Fig. 8a). (ii) Once a light-driven charge separation event occurs within PSI, oxidized Fd approaches PSI to bind at the stromal side for receiving the excited electron. Fine structural alterations within PSI triggered by the binding of Fd are transmitted to the luminal side for recruiting Cyt $c_6$ to the non-productive binding site (Fig. 8b). Increasing the affinity of non-productive binding sites for Cyt $c_6$ may protect the PSI reaction center from over-reduction when $F_A$ and $F_B$ are in a reduced state. (iii) PSI reduces Fd (Note, in our structure Ga-Fd cannot be reduced and thus Fd remains bound to PSI and Cyt $c_6$ remains bound at the non-productive site (Fig. 8c)). (iv) Once Fd is reduced by an excited electron and then detaches from PSI, Cyt $c_6$ will migrate to the P700 proximal productive binding site for subsequent electron transfer (Fig. 8d). (v) Finally, oxidized Cyt $c_6$ detaches from PSI, and another round of electron transfer reactions begins.

In summary, we propose that PSI engages in two types of Pc- or Cyt $c_6$-binding mechanism to the reaction center. One mechanism involves "pre-binding" of Pc to the N-terminal extension of PsaF as observed for *Chlamydomonas reinhardtii*. The alternative mechanism involves Fd-binding induced "non-productive" binding to facilitate efficient electron transfer in cyanobacteria. These recruiting mechanisms may have evolved further to provide the basis for efficient and robust electron transfer to P700 in green algae and vascular plants.

## Methods

**PSI trimer purification**. Thermophilic cyanobacteria of the *Thermosynechococcus elongatus* BP-1 mutant strain, genetically modified with an N-terminal His$_{10}$-tag on the PsaF subunit as well as a chloramphenicol-resistant gene as a selectable marker, were cultured under illumination for the purification of both PSI trimer and Cyt $c_6$. Cultivation was performed under continuous illumination at 30 μM photons m$^{-2}$ s$^{-1}$ at 50 °C in BG11 medium as described previously[61]. A total of 16 L culture was harvested when the OD$_{730\,nm}$ absorbance reached a value of one. His-tagged PSI trimers were purified from thylakoid membranes using the methodology described by Kubota et al.[43] with minor modifications as described below.

Harvested cells were concentrated by centrifugation (8000 × *g*, 2 min at 4 °C, JLA 9.1000 rotor, Avanti™ HP-26XP), resuspended in Buffer A containing 50 mM MES (pH 6.5), 25% glycerol, 20 mM CaCl$_2$ and 20 mM MgCl$_2$, then frozen for storage at −80 °C or used immediately for cell disruption as previously described[62]. Cells were disrupted in freshly prepared pre-cooled Buffer B (same components as in Buffer A with the addition of 1 mM benzamidine, 1 mM 6-aminohexanoic acid, 1 mM PMSF solubilized in ethanol, and 1 mM DNase I) and thylakoid membranes were separated from soluble cellular contents by ultracentrifugation (40,000 × *g*, 30 min at 4 °C, P50AT2 rotor, HITACHI himac CP80WX).

Thylakoid membranes were resuspended in Buffer A at a chlorophyll concentration of 1 mg/mL, and subsequently, a 10% β-DDM stock solution was added dropwise until the concentration of β-DDM reached 1%. The mixture was then incubated in the dark at 4 °C with gentle stirring for 45 min. After membrane solubilization, insoluble contents were removed by ultracentrifugation (50,000 × *g*, 30 min at 4 °C, P50AT2 rotor, HITACHI himac CP80WX) and the supernatant was applied to an open column with Ni-NTA Sepharose (Qiagen, Hilden, Germany). PSI trimers were eluted from the column using Buffer C containing 20 mM MES (pH 6.5), 20 mM CaCl$_2$, 20 mM MgCl$_2$, 0.05% β-DDM, and 100 mM imidazole. Eluted PSI trimers were concentrated by ultrafiltration at 3000 × *g*, 4 °C (Amicon15 ultrafiltration tubes, 100 kDa MWCO; Amicon, Miami, FL, USA) to a final chlorophyll concentration of around 10 mg/mL.

The GraDeR[34] approach was applied for exchanging the detergent β-DDM to glyco-diosgenin (GDN) and for the removal of excess free detergent. To achieve this a sucrose step gradient was prepared with sucrose concentrations of 1.7 M, 1.3 M, 0.9 M, 0.5 M, and 0.1 M from bottom-to-top, and GDN of 0.1%, 0.05%, 0.03% 0.01%, and 0.005% from bottom-to-top. After sucrose density gradient centrifugation (25,000 × *g*, 20 h at 4 °C, P28S rotor, HITACHI himac CP80WX, result see Supplementary Fig. 1a), PSI trimers from the corresponding band were

collected, and a desalting column (PD-10, GE Healthcare, Chicago, IL, USA) was used for removing sucrose and for buffer exchange. Final PSI trimers were concentrated to a chlorophyll concentration of 10 mg/mL in buffer D (10 mM Tricine, pH 8.0, 10 mM MgCl₂, and 0.005% GDN) and examined for purity, stability, and monodispersity using negative stain electron microscopy as described below. The final PSI trimer sample was flash frozen in liquid nitrogen in 10 µL aliquots using photophobic EP tubes and stored until further use at −80 °C. Protein concentration of PSI trimer was determined based on the chlorophyll content ratio of $C_{Pro}/C_{Chl} = 3.92$.

**Negative stain EM**. The quality of PSI protein samples was assessed by negative staining. An aliquot of 3.5 µL of the final sample was diluted 1/100 and applied to glow-discharged (5 mA, 10 s, Eiko) continuous carbon film coated copper grids (Nisshin EM, Tokyo, Japan). Staining was performed by applying 3.5 µL 2% uranyl acetate solution. After incubation for 30 s, staining solution was blotted using filter paper (Whatman #1; Whatman, Little Chalfont, UK) and dried at room temperature. EM grids were inspected using a H-7650 HITACHI transmission electron microscope at 80 kV acceleration voltage equipped with a 1× 1K Tietz FastScan-F114 CCD camera. A typical negative stain micrograph of trimeric PSI particles is presented in Supplementary Fig. 1b.

**Cyt c₆ purification**. Supernatant obtained by disruption of *T. elongatus* BP-1 cells and ultracentrifugation as described above was utilized for the isolation of Cyt $c_6$ based on a previous purification method[63] with some modifications. Solid ammonium sulfate (AS) was gradually added at 25 °C to the supernatant until saturation reached 45%. After stirring for 45 min at 25 °C the precipitate was removed by centrifugation (9500 × g, 30 min at 25 °C, JLA 9.1000 rotor, Avanti™ HP-26XP). The resulting supernatant was loaded onto a hydrophobic interaction chromatography column (HiTrap® Phenyl HP) pre-equilibrated with buffer containing 10 mM Tricine (pH 7.5) and 45% AS using a Fast Protein Liquid Chromatography system (FPLC, ÄKTA explorer). Fractions containing $c_6$ were eluted by decreasing the AS concentration to around 20%. After column-mediated desalting (PD-10, GE Healthcare), fractions were exchanged to a buffer containing 10 mM Tricine (pH 7.5) and 10 mM NaCl, and loaded onto an anion exchange column (HiTrap® Q HP) pre-equilibrated with the same buffer. Bound protein was eluted by increasing the NaCl concentration to around 100 mM and fractions with a pink coloration were pooled. The pooled sample was then subjected to a final gel filtration chromatography step (Superose® 75 16/60) using a running buffer containing 30 mM Tricine (pH 8.0) and 10 mM MgCl₂. Sample quality and redox state was assessed by measuring the $OD_{553nm}/OD_{273nm}$ absorbance ratio based on a previously published protocol[64] and absorbance properties were characterized by UV–Vis absorbance spectroscopy. The deep pink color and characteristic absorption peak at 553 nm verified the reduced state of purified Cyt $c_6$ when compared with previous spectroscopic results[65] of reduced Cyt $c_6$. SDS-PAGE analysis of purified Cyt $c_6$ is presented in Supplementary Fig. 1c.

**Ferredoxin purification and preparation of gallium-substituted ferredoxin**. A DNA fragment encoding native *T. elongatus* BP-1 ferredoxin (Fd) was cloned into the pET28a vector (Novagen, Madison, WI, USA) using *NcoI*−*Bam*HI restriction sites and expressed using the *Escherichia coli* strain BL21(DE3) cultured in Luria-Bertani (LB) medium. Purification of Fd and the subsequent substitution of its native [2Fe-2S] cluster with the redox inert [2Ga-2S] cluster were performed as previously described[27]. In brief, cells were harvested by centrifugation, resuspended in a buffer containing 50 mM Tris (pH 7.5) and 10% glycerol. After cell disruption by sonication resulting lysates were centrifuged for 30 min at 45,000 rpm, 4 °C (P45AT rotor, Hitachi CP80WX) and the resulting supernatant loaded onto an anion exchange cellulose column (DE52, Whatman). Red-colored fractions containing Fd were eluted using a buffer containing 50 mM Tris (pH 7.5) and 500 mM NaCl and dialyzed overnight at 4 °C against 3 L of buffer containing 50 mM Tris (pH 7.5) and 50 mM NaCl. The dialysate was then applied to a HiTrap Q HP column (GE Healthcare) in buffer containing 50 mM Tris (pH 7.5), and eluted using a linear NaCl gradient (0–1 M NaCl). Subsequently, for ammonium sulfate precipitation a 100% saturated AS buffer (50 mM Tris (pH 7.5) was added 1:1 to the pooled fractions. The precipitate was spun down (8000 × g, TOMY) and discarded. The sample was finally subjected to hydrophobic interaction chromatography using a phenyl Sepharose column (GE Healthcare) equilibrated in buffer A (50 mM Tris pH 7.5, 40% saturated ammonium sulfate). Elution was performed using a linear gradient of 0–100% buffer B (50 mM Tris pH 7.5) and red-colored fractions were collected.

For Gallium substitution, 45 mg of Fd was denatured by adding hydrogen chloride (HCl) at a concentration of 6 M to a final concentration of 1 M HCl. The turbid solution was pelleted (10 min, 10,000 × g at R.T., TOMY's centrifuge machine) and the white precipitate immediately rinsed with Milli-Q water, followed by resuspension in a buffer containing 100 mM Tris (pH 8.0). The above steps were repeated twice to ensure removal of all iron atoms from the Fd. The final protein precipitate was resuspended in a buffer containing 100 mM Tris (pH 8.0), 6 M guanidine hydrochloride, and 10 mM dithiothreitol in an anaerobic environment.

The denatured Fd was refolded at 4 °C by diluting it into a refolding buffer (2 mM GaCl₃, 2 mM Na₂S, 2 mM DTT, and 20 mM Tris, pH 8.0), and incubated at 4 °C overnight under anaerobic conditions. Refolded Ga-Fd was purified via HiTrap-Q column (GE Healthcare) chromatography using an elution gradient from 0 to 1 M NaCl in 20 mM Tris-HCl (pH 7.5). Elution profiles of the proteins were monitored by absorbance at 280 nm. Eluted fractions containing Ga-Fd were collected and concentrated by ultrafiltration using a 10 kDa cut-off Amicon Ultra-15 unit that was centrifuged in a swing out rotor (2000 × g, 4 °C). The concentrated sample was then loaded onto a Superdex 75 16/60 column (GE Healthcare) pre-equilibrated in 100 mM Tris (pH 8.0) at a flow rate of 0.5 ml/min in the same buffer. Fractions containing Ga-Fd were detected by absorbance at 280 nm and then subjected to a final size exclusion chromatography step. The concentration of native Fd and Ga-Fd were calculated from their molar extinction coefficients of $\varepsilon_{422} = 9.68$ mM/cm and $\varepsilon_{280} = 170.2$ mM/cm, respectively[27]. The purity of Ga-Fd was assessed by SDS-PAGE analysis as shown in Supplementary Fig. 1d.

**ITC measurements**. All ITC measurements were performed with a MicroCal PEAQ-ITC instrument (Malvern Panalytical, UK) at 25 °C. The concentrations of Ga-Fd in the syringe and PSI trimer in the cell were 600 µM and 11 µM, respectively. All protein solutions were subjected to buffer exchange into 30 mM Tricine-NaOH buffer (pH 8.0) containing 10 mM MgCl₂, 50 mM NaCl, and 0.005% GDN by using PD-10 columns (GE Healthcare Life Sciences), and air bubbles were removed by centrifugation for 5 min at 10,000 × g prior to ITC. The following ITC parameters were used: titration, 19 injections; initial delay, 60 s; spacing time, 180 s; reference power, 10 µcal/s; and stirring speed, 750 rpm. The injection volume was 0.4 µL for the first injection and 2 µL for the remaining injections. The heat of dilution was measured by titrating 600 µM Ga-substituted Fd in the syringe into a sample cell filled with buffer alone. The heat flow and binding isotherm were calculated by subtracting the heat of dilution. The data were fitted to the one set of sites binding model incorporated in MicroCal PEAQ-ITC analysis software. Thermodynamic parameters are shown as mean ± S.E.M. derived from three independent ITC experiments.

**Cryo-grid preparation**. A total of 10 µL of purified PSI trimer at a protein concentration of 37 µM (40 mg/mL), 1 µL of purified Ga-Fd in concentration of 400 µM (4.4 mg/mL), and 2 µL of purified Cyt $c_6$ in concentration of 75 µM (0.75 mg/mL) were mixed together at ice temperature to a final concentration of 30 mg/mL PSI trimer. After incubation on ice in the dark for 2 h, an aliquot of 2.6 µL was applied to a glow-discharged (JEC-3000 FC, 30 s, carbon support film facing up) Quantifoil cryo-EM grid (R 1.2/1.3 Cu 300 mesh) and plunge frozen in liquid ethane using a Vitrobot Mark IV (Thermo Fisher Scientific, Waltham, MA, USA) fitted with Whatman #1 filter paper for blotting at 4 °C and 100% humidity for 3.5 s at blot force 0. Frozen grids were transferred to liquid nitrogen for storage. Screening was performed using a Talos Arctica (Thermo Fisher Scientific) transmission cryo-electron microscope operated at 200 kV. Cryo-grids exhibiting appropriate particle distribution and ice thickness were taken out of the Talos Arctica and further used for high-resolution data collection.

**Cryo-EM image data collection**. Using the auto-grid adapter cartridge (JEOL, Tokyo, Japan), a single pre-screened cryo-grid clipped as an auto-grid (Thermo Fisher Scientific) was transferred to a CRYO ARM 300 (JEOL) transmission cryo-electron microscope operated at 300 kV and equipped with a cold-field emission gun and an in-column Ω filter. Image acquisition was performed using flood beam parallel illumination in bright field imaging mode. Movies were automatically recorded by SerialEM using image shift (5 × 5 holes per stage position) and a K3 Direct Detection Camera (Gatan, AMETEK, Pleasanton, CA, USA) in CDS mode at a nominal magnification of ×60,000 at the camera level, corresponding to a pixel size of 0.806 Å with 48 frames in 3 s exposure time, resulting in a total dose of 48 e⁻/Å². A total of 3018 movies were collected in series within a defocus range of −0.5 µm to −1.5 µm. A typical micrograph and angular distribution of protein complexes are shown in Supplementary Figs. 2b and 3b, respectively.

**Cryo-EM image processing**. All image processing was performed using RELION 3.1 and UCSF CHIMERA[66] on a local GPU workstation equipped with two GPUs (see Supplementary Fig. 2a for the workflow). A total of 3018 movies were divided into six groups, and dose-weighted motion-correction was performed by the in Relion implemented MotionCor2[67] algorithm. For each motion-corrected movie, the contrast transfer function (CTF) was estimated using CTFFIND4[68]. After CTF estimation, 1959 movies with an estimated resolution of better than 5 Å and good Thon rings (Supplementary Fig. 2c) were kept for further processing. For de novo building of an initial model, 40,483 particles were automatically picked from one of the 500 movie groups using the Laplacian of Gaussian method. Picked particles were subjected to one round of reference-free 2D classification and the initial model was calculated using 30,580 particles of the best 2D classes. Supplementary Fig. 2e presents an isosurface visualization of the initial model at four different thresholds. Auto-picking on movies of all six groups was performed by using the initial model as a 3D reference. A total of 511,737 particles were picked from all movies and 2D classification was performed on the particles picked for each group of movies. Good 2D classes (Supplementary Fig. 2d) containing a total of 366,174

particles were selected, pooled, and used for the 3D classification. Particles of the best 3D class, indicated by a red caption in Supplementary Fig. 2f representing 56.6% (207,142 particles) of all particles, were selected for 3D reconstruction using standard auto-refinement. Subsequently, several rounds of CTF refinement and Bayesian polishing were performed until the overall map resolution did not improve any further. Following standard post-processing procedures, the masked final map gave an estimated resolution of 2.06 Å without symmetry applied (C1) and 1.97 Å with symmetry applied (C3). No obvious difference could be observed between the C1 and C3 maps in terms of structural features. The C3 symmetrized map was used for model building because it exhibited better Euler angle distribution and a reduced level of noise. Local resolution distribution (Supplementary Fig. 4a) of the C3 symmetrized map was calculated using RELION.

**Atomic model building and refinement.** Coot[69], Phenix, MolProbity[70], and UCSF Chimera[66] were used for model building and real-space refinement. The X-ray crystal structures of the *Synechococcus elongatus* PSI trimer determined at 2.5 Å resolution (PDB ID: 1JB0[4]) and of the *T. elongatus* BP-1 ferredoxin X-ray crystal structure determined at 1.5 Å resolution (PDB ID: 5AUI[27]) were used as a starting model for real-space refinement in Phenix 1.19.2 after manual fitting in UCSF Chimera 1.13.1. Coot 0.9.5 was used for manual correction and minimization of rotamer, geometry, and Ramachandran outliers in the real space refined model. Accurate non-metal ligand restraint files were downloaded from Grade Server (http://grade.globalphasing.org) and utilized during refinement. To account for the observed presence of magnesium on both sides of the chlorine ring plane the chirality limitation of the magnesium binding plane in the restraints of chlorophylls (CLA and CL0) was deleted to allow reasonable fitting within the density map. In addition, new iron–sulfur cluster restraints[71] using a rhomboid geometry were applied for iron–sulfur cluster (SF4) refinement. The newly modified restraint files have been uploaded together with the model to the wwPDB deposition site Visualization of maps and models in all figures was performed using PyMOL 2.3.2, UCSF Chimera 1.13.1 and ChimeraX[72] 1.2.5. For the positioning of Cyt $c_6$ presented in Fig. 2, Phenix "Dock in map" routine (https://phenix-online.org/documentation/reference/dock_in_map.html) was performed with the input of corresponding local density map and the *T. elongatus* BP-1 Cyt $c_6$ X-ray crystal structure determined at 1.7 Å resolution (PDB ID: 6TR1[65]).

**Statistics and reproducibility.** No statistical method was applied to presuppose the protein sample size or shape, and the experiments were not randomized. The investigators were not blinded to allocation during experiments and outcome assessment. The purification experiments of PSI trimers, ferredoxin and Cytochrome $c_6$ were repeated over three times, which presented almost identical sample features. The ITC measurements were repeated three times and similar results were obtained (Supplementary Data 1 shows the source data of ITC measurement described in the paper). Cryo-EM data were collected from a single cryo-grid. Individual images with estimated resolution of worse than 5 Å and blurry Thon rings as well as unsatisfying ice thickness were excluded from the data set manually. Detailed statistics of data collection, processing, and refinement statistics are summarized in Supplementary Table 1.

**Reporting summary.** Further information on research design is available in the Nature Research Reporting Summary linked to this article.

## Data availability
All 3018 raw cryo-EM movies were deposited in the Electron Microscopy Public Image Archive (EMPIAR) under the accession number EMPIAR-10928. The final cryo-EM density map has been deposited in the Electron Microscopy Data Bank (EMDB) under the accession number EMD-31605. The atomic coordinates of the refined PSI:Fd model have been deposited in the Protein Data Bank (PDB) under the accession number 7FIX. All relevant data are available from the corresponding authors upon reasonable request.

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

## Acknowledgements

We are grateful for the technical support and valuable scientific input for this project by Hisako Kubota-Kawai, Yuko Misumi, Hideaki Tanaka, Mika Hirose, Norbert Krauß, Orkun Çoruh, and Takayuki Kato. This work was funded by the JST-CREST under grant number JPMJCR20E1 (G.K.) and a Grants-in-Aid for Scientific Research under grant number 21H02417 (G.K.) from MEXT-KAKENHI, the Platform Project for Supporting Drug Discovery and Life Science Research from AMED under grant number JP20am0101117 (K.N.) and JP16K07266 (C.G.), and JP22ama121001j0001 to Masaki Yamamoto, C.G. and G.K. and the Cyclic Innovation for Clinical Empowerment (CiCLE) grant number JP17pc0101020 from AMED to K.N. and G.K., and National Research Foundation of Korea (NRF) grants funded by the Korean government under grant number 2019R1A2C1004954, National Research Council of Science & Technology (NST) grant funded by the Korean government under grant number CCL22061-100 and the KBSI fund under grant number C220000, C230130, C280320, and C270100 to Y.-H.L., and the Deutsche Forschungsgemeinschaft (DFG) within the framework of the Research Training Group 234 'MiCon' (M.M.N). This work is also supported by the scholarship from China Scholarship Council (CSC) for J.L.'s study at Osaka University (CSC No. 201706220064).

## Author contributions

G.K., M.R., and M.M.N. initiated the study; G.K. and C.G. designed research; J.L., N.H., F.M., A.K., and Y.L. performed experiments; J.L., N.H., Y.L., Y.-H.L., and C.G. analyzed the data; K.N. supervised research and J.L., C.G., and G.K. wrote the manuscript with input and comments from all authors.

## Competing interests

The authors declare no competing interests.
