## [Peer Review File · Communications Biology]

This manuscript has been previously reviewed at another Nature Portfolio journal. This document only contains reviewer comments and rebuttal letters for versions considered at Communications Biology.

REVIEWERS' COMMENTS:

Reviewer #1 (Remarks to the Author):

The work reports a very high-resolution structure of PSI in complex with Fd and a loosely bound Cyt c6. Although PSI-Fd structures from different sources have been reported previously, the current work extends the earlier low-resolution structures to a higher resolution, which provided detailed information about the interaction between PSI and Fd, and thus is important for understanding the assembly and the electron transfer mechanism of this complex. In addition, the authors also suggested a non-productive binding site of Cyt c6 in PSI based on their structure. Overall, this is a nice work providing novel information. The work is solid in structural analysis, and the manuscript is well performed and clearly written.

The analysis of the structural changes upon the formation of the PSI-Fd complex is nice. However, since structures of both free PSI and free Fd are crystal structures, the authors need to distinguish between the different conformations induced by complex formation and those caused by crystal packing.

Figures should be mentioned in the text by their number. Now the Figure 7 was cited before Figure 4 in the maintext.

August 23, 2022

Reviewer #1 (Remarks to the Author):

The work reports a very high-resolution structure of PSI in complex with Fd and a loosely bound Cyt c6. Although PSI-Fd structures from different sources have been reported previously, the current work extends the earlier low-resolution structures to a higher resolution, which provided detailed information about the interaction between PSI and Fd, and thus is important for understanding the assembly and the electron transfer mechanism of this complex. In addition, the authors also suggested a non-productive binding site of Cyt c6 in PSI based on their structure. Overall, this is a nice work providing novel information. The work is solid in structural analysis, and the manuscript is well performed and clearly written.

Thank you very much for finding our manuscript of a PSI:Fd:Cyt c₆ triple complex significant.

The analysis of the structural changes upon the formation of the PSI-Fd complex is nice. However, since structures of both free PSI and free Fd are crystal structures, the authors need to distinguish

between the different conformations induced by complex formation and those caused by crystal packing.

We have already thought the same way, so we examined the structural changes only for C α -carbon basis because most of the crystal packings were mediated by side chains of amino acids. In order to alleviate your worries, we revised corresponding part of the paragraph of **Structural and thermodynamic changes upon FSI:Fd complex formation** by inserting one sentence as follows; “Because individual side chains may be involved in the crystal packing for X-ray structures of PSI or Fd, we intentionally used the α -carbon positions of main chain structures for comparisons and the C α -baton models without side chains were shown in Fig. 6.”.

Figures should be mentioned in the text by their number. Now the Figure 7 was cited before Figure 4 in the maintext.

We are very sorry for careless mistake. It was a typo of wrong figure number. Now it is correctly cited in numerical order.